# STRUCTURED DEEP FACTORIZATION MACHINE: TOWARDS GENERAL-PURPOSE ARCHITECTURES

## ABSTRACT

In spite of their great success, traditional factorization algorithms typically do not support features (e.g., Matrix Factorization), or their complexity scales quadratically with the number of features (e.g, Factorization Machine). On the other hand, neural methods allow large feature sets, but are often designed for a specific application. We propose novel deep factorization methods that allow efficient and flexible feature representation. For example, we enable describing items with natural language with complexity linear to the vocabulary size—this enables prediction for unseen items and avoids the cold start problem. We show that our architecture can generalize some previously published single-purpose neural architectures. Our experiments suggest improved training times and accuracy compared to shallow methods.

## 1 INTRODUCTION

In recent years, predictive tasks that traditionally have been solved with factorization are now being studied within the context of neural networks. These solutions often work as black boxes, and many times they are designed specifically for a single task with an arbitrary network that may not have much justification. We propose Deep Structured Factorization Machine, a family of general-purpose factorization techniques that can be used stand-alone or as a "design pattern" within a larger neural network. Our work provides some insight into how to enable general-purpose factorization within neural architectures without losing interpretability and a principled design.

Previous factorization methods do not scale to large feature sets and make strong assumptions about their latent structure. Our main contribution is that we enable a general-purpose framework that enables efficient factorization of datasets with complex feature sets. For example, applications of factorization in natural language scale quadratically in the number of words in the vocabulary. Our solution allows inference with linear runtime complexity on the vocabulary size. Previous work has explored how to improve factorization's accuracy (see § 3.3) with its current limitations withstanding; alternatively, some have proposed how to make it tractable for a particular domain—for example, text (Zheng et al., 2017). We believe that we are the first ones to propose an efficient general-purpose method. Interestingly, our experiments indicate that Structured Deep Factorization has large improvements in predictive accuracy and runtime compared to some recent ad-hoc models.

## 2 PRELIMINARIES

Factorization Machine (Rendle, 2010) is one of the most succesful methods for general purpose factorization. Rendle (2010) formulated it as an extension to polynomial regression. Consider a degree-2 polynomial (quadratic) regression, where we want to predict a target variable $y$ from a feature vector $\mathbf{x} \in \mathbb{R}^n$:

$$\hat{y}(\mathbf{x}; \mathbf{b}, \mathbf{w}) = \omega\big(b_0 + \lambda^1(\mathbf{x}, \mathbf{b}) + \lambda^{\mathrm{p}}(\mathbf{x}, \mathbf{w})\big) \tag{1}$$

Here, $\lambda^1$ and $\lambda^p$ are one-way and pairwise interactions:

$$\lambda^1(\mathbf{x}, \mathbf{b}) \triangleq \sum_i b_i x_i \tag{2}$$

$$\lambda^p(\mathbf{x}, \mathbf{w}) \triangleq \sum_{i=1}^n \sum_{j=i+1}^n w_{i,j} \; x_i x_j \tag{3}$$

In words, $n$ is the total number of features, the term $b_0$ is an intercept, $b_i$ is the strength of the $i$-th feature, and $w_{i,j}$ is the interaction coefficient between the $i$-th and $j$-th feature. The function $\omega$ is an activation. Choices for $\omega$ include a linear link ($\omega(x) = x$) for continuous outputs, or logistic link ($\omega(x) = \log\left(\frac{\exp(x)}{\exp(x+1)}\right)$) for binary outputs.

Factorization Machine replaces the two-way individual pairwise parameters with shared parameters $\boldsymbol{\beta}_i$. This is a rank-$r$ vector of factors—*embeddings* in the neural literature—that encode the interaction between features:

$$\lambda^{\mathrm{FM}}(\mathbf{x}, \boldsymbol{\beta}) \triangleq \sum_{i=1}^n \sum_{j=i+1}^n x_i \boldsymbol{\beta}_i \cdot x_j \boldsymbol{\beta}_j \tag{4}$$

Intuitively, the dot product ($\cdot$) returns a scalar that measures the (dis)similarity between the two factors. Polynomial regression has $n^2$ interaction parameters, and Factorization Machine has $n \times r$. While using $r \ll n$ makes the model less expressive, this is often desirable because factorization is typically used when features have some shared latent structure. Factorization Machine may dramatically reduce the number of parameters to estimate; however, inference has runtime that scales quadratically with the number of features—it requires $\mathcal{O}(n^2)$ dot products. Rendle (2010) shows that when the feature vector $\mathbf{x}$ consists only of two categorical features in one-hot encoding, Factorization Machine is equivalent to the popular Matrix Factorization algorithm (Koren et al., 2009).

## 3 STRUCTURED FACTORIZATION MACHINE

Factorization Machine has the following working-assumptions and limitations:

1. **Strong parameter sharing**. Factorization Machine assumes that all of the pairwise interactions factorize into a common shared space. This may not be the case when using large and complex feature sets. To the extent of our knowledge, softening this assumption has not been studied in the literature before. We study how to do so in § 3.1.

2. **Intractable for large feature sets**. Doing inference on Factorization Machine has a runtime complexity that scales quadratically with the number of features. Thus, it is intractable for complex feature sets such as text. We address this in § 3.2.

We propose Structured Factorization Machine as a building block that we will extend with neural methods. It is a simple yet effective way to model structure between groups of features. In Factorization Machine, all the features interact with each other. In the structured model, the features only interact with features from a different group. We do this by defining groups of features $\boldsymbol{\kappa}$. We define Structured Factorization Machine as:

$$\hat{y}(\mathbf{x}; \mathbf{b}, \boldsymbol{\beta}, \boldsymbol{\kappa}) = \omega\left(b_0 + \sum_{i=1}^n b_i x_i + \sum_{i=1}^{|\boldsymbol{\kappa}|} \sum_{j=i}^{|\boldsymbol{\kappa}|} \lambda^s(\mathbf{x}, \boldsymbol{\beta}, \boldsymbol{\kappa}_i, \boldsymbol{\kappa}_j)\right) \tag{5}$$

The interactions occur from features in different groups:

$$\lambda^s(\mathbf{x}, \boldsymbol{\beta}, \mathbf{I}, \mathbf{J}) \triangleq \sum_{i \in I} \sum_{j \in J} x_i \boldsymbol{\beta}_i \cdot x_j \boldsymbol{\beta}_j \tag{6}$$

For example, consider a model with four features ($n = 4$). If we define $\boldsymbol{\kappa} = \{\{1, 2\}, \{3, 4\}\}$, feature $x_1$ would only interact with $x_3$ and $x_4$. Without loss of generality, we could define a model that is equivalent to a shallow Factorization Machine by allowing each feature to be in a singleton group: $\boldsymbol{\kappa} = \{\{1\}, \{2\}, \{3\}, \{4\}\}$.

Figure 1 compares existing factorization methods with our novel models. In the rest of this section we review them.

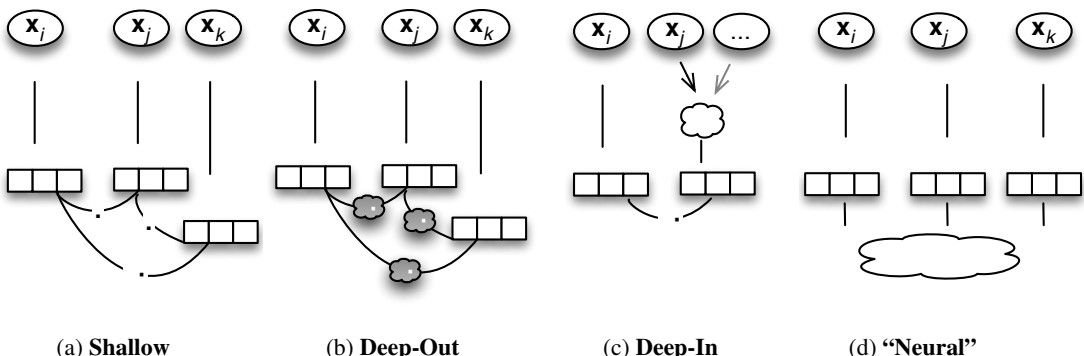

(a) **Shallow**          (b) **Deep-Out**          (c) **Deep-In**          (d) **"Neural"**

Figure 1: **Network architectures for factorization models**. The white clouds (☁) represent one or more factorized deep layers, and the dark clouds (☁) softens the dot product constraints.

## 3.1 Structured Deep-Out Factorization Machine

We now contrast Polynomial Regression (Equation 1) and Factorization Machine. Polynomial regression is convex, and thus can be optimized globally. Because Factorization Machine is non-convex, optimization can only guarantee to find local optima.

The factorization assumption is more desirable than Polynomial Regression when there is shared structure between the features. The larger number of parameters of Polynomial Regression makes it likelier to overfit. However, we consider two properties of independent interactions that are desirable. First, it is unclear if all interactions strongly factorize when using very large feature sets. For example, some interactions may have different importance—perhaps because some features are irrelevant and should be discarded. Second, it may be easier to optimize a model with individual parameters than a model with factorized parameters. Polynomial regression converges quickly because the likelihood function is convex—a unique solution exists.

We aim to combine the advantage of both models. We propose a hybrid factorized-independent method that we call Structured Deep-Out Factorization Machine, because it extends Structured Factorization Machine outside the dot product (see Figure 1b):

$$\omega\left(b_0 + \sum_{i=1}^n b_i x_i + \psi\big(\sum_{i=1}^{|\kappa|}\sum_{j=i}^{|\kappa|} w_{i,j}\,\lambda^s(\mathbf{x}, \boldsymbol{\beta}, \boldsymbol{\kappa}_i, \boldsymbol{\kappa}_j)\,\big)\right) \tag{7}$$

Here, the the embeddings $\boldsymbol{\beta}$ are factorized and shared across the pairwise interactions. The parameters $w$ are independent to each *interaction group* defined by $\boldsymbol{\kappa}$.

The function $\psi$ is any activation function. If we were interested in interpreting the parameters, we would constrain $w$ to be non-negative and choose $\psi$ to be linear. Alternatively, the Rectified Linear Unit (ReLU) activation may be desirable because it is easy to optimize (Krizhevsky et al., 2012): $\psi(x) = \max(0, x)$. We leave for future work the consideration of deeper networks.

## 3.2 Structured Deep-In Factorization Machine

Structured Deep-In Factorization Machine allows treating a group of features as a single one; it extracts features from each feature group, and builds latent factors from them. Consider Figure 1c: the first group only has a single feature which is projected to an embedding (just like a regular Factorization Machine); the second group has multiple features, which are together projected to a single embedding. More formally:

$$\omega\left(b_0 + \sum_{i=1}^n b_i x_i + \sum_{i=1}^{|\kappa|}\sum_{j=i}^{|\kappa|} \phi_i(\mathbf{x}_{\boldsymbol{\kappa}_i}) \cdot \phi_j(\mathbf{x}_{\boldsymbol{\kappa}_j})\right) \tag{8}$$

In this notation, $\mathbf{x}_{\boldsymbol{\kappa}_i}$ is a subvector that contains all of the features that belong to the group $\boldsymbol{\kappa}_i$. Thus, $x_{\boldsymbol{\kappa}_i} = [x_j | j \in \boldsymbol{\kappa}_i]$. The intuition is that by grouping (sub-)features as a single entity, we can can reason on a higher level of abstraction. Instead of individual sub-features interacting among each other, the entities interact with other entities. Here, $\phi_i$ is a feature extraction feature that inputs the $i$-th feature group of the instance, and returns an embedding. The simplest implementation for $\phi_i$ is a linear fully-connected layer, where the output of the $r$-th entry is:

$$\phi_i\big(\mathbf{x}_i; \boldsymbol{\beta}\big)_r = \sum_{a=1}^{d_i} \beta_{r_a} x_{i_a}$$

Within a group, a feature extraction function may allow for subfeatures to interact with each other. Across groups, entities interact with each other via the output of $\phi$ only.

**Leveraging item descriptions to circumvent the cold-start problem**    We now describe how we can use Structured Deep-In Factorization Machine for large feature sets—such as natural language. In this scenario Factorization Machine is not tractable: if we use each word in the vocabulary as independent feature, we have to consider the interaction of each word with every word in the vocabulary and the other features, which means the number of interaction terms would depend quadratically on the size of the vocabulary. A traditional work-around is to ignore the features, and use matrix factorization by using a unique one-hot index for each item (e.g, each image/text), which avoids the quadratic dependence on the vocabulary size, but in this case the model suffers from the cold-start issue: inference is not possible for a new item not seen during training, even if it shares many words with existing items.

We can use the Deep-In model to both use large feature sets and overcome the cold-start problem. This is only possible when there is an alternative description of the item available (for example an image or a text). In Figure 2, we show how we address this problem by treating the words as indexed features, but placed within a structured feature group $\boldsymbol{\kappa}_w$. A feature extraction function $\phi$ acts on the features in $\boldsymbol{\kappa}_w$, and the other features interact with the words only via the output of $\phi$. Notice that this implies we can precompute and store the latent factors of the labels seen during training, and predictions during inference can be sped-up. For example if we have two feature groups (e.g, a label and an item), first we compute the feature extraction function to the unseen items and their embeddings, and then we simply apply a dot product over the stored vectors of the labels.

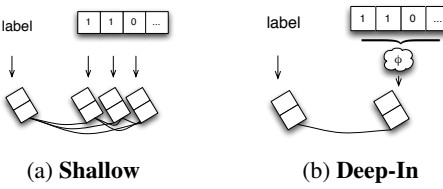

(a) **Shallow**          (b) **Deep-In**

Figure 2: Comparison of how factorization may use item descriptions features.

### 3.3    NEURAL FACTORIZATION

Figure 1d shows a Neural Factorization model that has been discovered multiple times (Dziugaite & Roy, 2015; Guo et al., 2017). It replaces the dot product of factors with a learned neural function, which has been shown to improve predictive accuracy for various tasks. Unlike our work, this method has the same drawbacks as regular Factorization Machine for feature types such as natural language. As discussed in §3.2, applying this method for text-based inputs would result in either the cold-start problem or quadratic dependence on the vocabulary. It would be straightforward to combine Neural Factorization with Structured Deep-In Factorization Machine, which is not explored in this work. Note that if the dot product is replaced with a neural function, fast inference for cold-start documents using pre-computed label embeddings is no longer possible.

### 3.4 LEARNING FROM DATA

We can learn the the parameters of a deep factorization model $\theta$ using training data by minimizing a loss function $\mathcal{L}$:

$$\arg\min_{\theta} \sum_x \mathcal{L}\big(y(x), \hat{y}(x; \theta)\big) + \gamma ||\theta||^{\beta} \tag{9}$$

Here, $y(x)$ is the true target value for $x$ obtained from training data, and $\hat{y}(x)$ is the one estimated by the model; the hyperparameter $\gamma$ controls the amount of regularization. For the labeling and classification tasks, we optimize the binary cross-entropy for $y \in \{0, 1\}$:

$$\mathcal{L}(y, \hat{y}) = -\big(y \log(\hat{y})\big) - (1 - y) \log(1 - \hat{y}) \tag{10}$$

For the regression tasks where the target value is continuous, we optimize the mean squared error (MSE):

$$\mathcal{L}(y, \hat{y}) = (y - \hat{y})^2 \tag{11}$$

Neural models are typically learned using mini-batch updates, where the incremental descent is performed with respect to several instances at a time. For the implementation of this paper, we built our models using the Keras programming toolkit (Chollet et al., 2015), that is now part of Tensorflow (Abadi et al., 2015). It enables automatic differentiation, and is bundled with a general-purpose optimization algorithm called ADAM (Kingma & Ba, 2014) that needs no tuning of gradient step sizes.

For the multi-label classification tasks, we use Structured Factorization Machine with a binary output. In this case we would have at least two feature groups—one of the feature groups is the label that we want to predict, and the other group(s) is the input from which we want to make the prediction. The output indicates whether the label is associated with the input ($y = +1$), or not ($y = 0$). The datasets we use for our labeling experiments only contains positive labels, thus for each training example we sample a set of negative labels equal to the number of positive labels. We choose one of the following sampling strategies according to the best validation error, in each case excluding the actual positive labels for each training example – (i) uniformly from all possible labels, or (ii) from the empirical distributions of positive labels. Other sampling strategies have been proposed (Rendle et al., 2009; Rendle & Freudenthaler, 2014; Anonymous, In submission).

## 4 EMPIRICAL RESULTS

We now address our working hypotheses. For all our experiments we define a development set and a single test set which is 10% of the dataset, and a part of the development set is used for early stopping or validating hyper-parameters. Since these datasets are large and require significant time to train on an Nvidia K80 GPU cluster, we report results on only a single training-test split. For the labeling and classification tasks we use evaluation metric called Area Under the Curve (AUC) of the Receiver Operating Characteristic (ROC). Since we only observe positive labels, for such tasks in the test set we sample a labels according to the label frequency. This ensures that if a model merely predicts the labels according to their popularity, it would have an AUC of 0.5. A caveat of our evaluation strategy is that we could be underestimating the performance of our models—there is a small probability that the sampled negatives labels are false negatives. However, since we apply the evaluation strategy consistently across our methods and baselines, the relative difference of the AUC is meaningful. We choose the AUC as a metric because it is popular for both classification and ranking problems. For the regression tasks, we use MSE as the evaluation metric. In preliminary experiments we noticed that regularization slows down convergence with no gains in prediction accuracy, so we avoid overfitting only by using early stopping. We share most of the code for the experiments online[1] for reproducibility.

### 4.1 IS THE FACTORIZATION HYPOTHESIS TOO STRONG FOR LARGE FEATURE SETS?

Factorization algorithms are desirable for smaller feature sets (e.g, users and items) that have shared latent structure, because a more parsimonious model is less prone to overfitting. We now investigate

---

[1] https://goo.gl/zEQBiA

if relaxing the factorization constraints for models with larger and more heterogeneous features is useful. We do this by comparing deep-out and a shallow structured models under the same conditions. For reproducibility details on our grid-search see Appendix A.1. We test this on classification, regression, and collaborative filtering (with implicit feedback prediction) tasks, respectively:

- **Subscription prediction** We predict the binary outcome of whether a marketing campaign is successful using the UCI Bank Marketing Data Set (Lichman, 2013; Moro et al., 2014). This is a small dataset with only 45,211 observations. We use 17 categorical and real-valued features.

- **Airline delays** We predict the real-valued delay of american flights using the 2008 RITA dataset[2]. using only 8 categorical features. This dataset is 7 million rows.

- **Course prediction** On this collaborative filtering task with implicit feedback, we use partial study plans collected from a company that has an educational-technology presence. We only consider data from students with three or more course enrollments, and courses that have had at least enrollments from 10 students. The resulting dataset contains approximately 36 million enrollment instances from roughly 4.7 million students and 2,578 courses (out of 2,930 that exist on the ontology). For sampling implicit feedback, we sample the course and subject variables together. The features we use are all categorical: student id, course id, subject level 1, subject level 2, subject level 3, university id, student year (discretized), and data source types.

For both shallow and deep models, we define the groups the same way: we group each continuous feature by itself; we group the multiple dimensions of the one-hot encoding of categorical variables together. This way, for example, the 12 dimensions that are used to encode a "month" feature don't interact with each other. Table 1 summarizes our results. Adding a handful of deep-out parameters is very effective for improving forecasting quality or training time. For example, shallow factorization does not do much better than random chance in the subscription prediction, but the deep-out approach improves the AUC by roughly 35%. We hypothesize that this dataset has features with little shared structured. On the courses dataset, the deep method is almost twice as fast as the shallow method, with some modest improvement on performance.

Table 1: Results of softening the factorization assumption. The reported time is the number of minutes it takes to train all of the hyper-parameter combinations (except for the course dataset that we only report the best model because the grid-search is adaptive)

| Dataset | Shallow | | Deep Out | |
| --- | --- | --- | --- | --- |
| | Performance | Time (mins) | Performance | Time (mins) |
| Subscription (AUC) | .54 | **1.2** | **.72** | 1.8 |
| Airline (RMSE) | 1,349.47 | 177.2 | **1,337.0** | **38.7** |
| Courses (AUC) | .79 | 98* | **.80** | **54.9*** |

## 4.2 STRUCTURED DEEP-IN FACTORIZATION MACHINE

In this section, we focus on natural language processing tasks. Table 2 provides descriptive statistics of the datasets that we consider. To choose the feature extraction function $\phi$ for text, we use a Convolutional Neural Network (CNN) that has been shown to be effective for natural language tasks (Kalchbrenner et al., 2014; Weston et al., 2014). In Appendix A.2 we describe this network and its hyper-parameters. Instead of tuning the hyper-parameters, we follow previously published guidelines (Zhang & Wallace, 2015).

### 4.2.1 CAN DEEP-IN FM BE USED TO ESTIMATE ITEM EMBEDDINGS FROM TEXT?

We experiment on two datasets to predict a label from text, obtained from an educational technology company:

---

[2]http://stat-computing.org/dataexpo/2009/the-data.html

Table 2: Details of labelling task datasets

|  | Number of documents | Total number of labels | Mean labels per document | Mean document length (words) | Negative label sampling strategy |
|---|---|---|---|---|---|
| Concepts | 42,710 | 186,215 | 117 | 2,117 | Uniform |
| Skills | 544,388 | 25,299 | 72 | 276 | Empirical |
| CiteULike users | 16,980 | 5,551 | 12 | 195 | Empirical |

1. **Concept labeling.** This dataset contains passages of texts related to topics taught at university courses. Subject matter experts annotated them with concepts. Our task is to design a model that can infer concepts from text.

2. **Skill labeling.** This dataset contains job posting texts annotated with the skills that are deemed to be useful by hiring managers. Thus, we try to predict the skills required for a job, particularly those that are not explicitly mentioned in the job posting or description.

For each task, we build two feature groups: the labels and the document text. The document texts are represented as a sequence of one-hot encoded words. As we describe in § 3.2, it is reasonable to use Matrix Factorization and ignore the item descriptions. Table 3 shows that Structured Deep-In Factorization Machine outperforms this strategy. Unfortunately, we cannot compare to Factorization Machine because it is intractable with a large vocabulary, as it would lead to interactions that scale quadratically in the vocabulary size (100,000 in our case).

### 4.2.2 IS DEEP-IN FM EFFECTIVE FOR COLD-START PREDICTIONS?

Matrix factorization cannot generalize to unseen items. Instead, we compare to Collaborative Topic Regression (CTR– Wang & Blei (2011)), a method with an open-source Python implementation[3] that leverages text features.

Because the implementation we found is relatively slow, we cannot test them on the concept and skill datasets. Instead, we use the **CiteULike** dataset which consists of pairs of scientific articles and the users who have added them to their personal libraries. We use it to predict users who may have added a given article to their library. We compare the performance of Structured Deep-In Factorization Machine with CTR using pre-defined cross-validation splits[4]. We use 1% of the training set for early stopping.

For CTR we use the hyper-parameters reported by the authors as best, except for $r$ which we found had a significant impact on training time . We only consider $r \in \{5, 10, 15\}$ and choose the value which gives the best performance for CTR (details in Appendix 6). On the warm-start condition, CTR has an AUC of 0.9356; however, it shows significant degradation in performance for unseen documents an it only performs slightly better than random chance with an AUC of 0.5047. On the other hand, Deep-In FM achieves AUC of 0.9401 on the warm-start condition, and it only degrades to 0.9124 on unseen documents. For completeness, we also tested Deep-In FM on cold-start prediction for the Concepts and Skills dataset, on 1% of documents unseen during training. We obtained AUC of 0.92 and 0.67, respectively, which are within 3% of the warm-start results. Deep-In FM can also be trained over ten times faster, since it can leverage GPUs.[5] We also note that we have not tuned the architecture or hyper-parameters of the feature extraction function $\phi$ for each dataset and greater improvements are possible by optimizing them.

### 4.2.3 COMPARISON WITH ALTERNATIVE CNN-BASED TEXT FACTORIZATION

We now compare with a method called DeepCoNN, a deep network specifically designed for incorporating text into matrix factorization (Zheng et al., 2017)—which reportedly, is the state of the

---

[3]https://github.com/arongdari/python-topic-model

[4]For warm-start we use `https://www.cs.cmu.edu/~chongw/data/citeulike/folds/cf-train-1-items.dat` and for cold-start predictions, we use the file `ofm-train-1-items.dat` and the corresponding test sets for each

[5]Deep-In FM and MF were trained on an Nvidia K80 GPU, while CTR was trained on a Xeon E5-2666 v3 CPU.

Table 3: Labeling results for warm-start documents

| Model | Concepts | | Skills | |
|---|---|---|---|---|
| | AUC | Train mins. | AUC | Train mins. |
| Deep-In F.M. | **0.92** | 1,876 | **0.65** | 3,256 |
| Matrix Fact. | 0.88 | **64** | 0.61 | **8** |

Table 4: Yelp rating prediction

| | MSE | Gain over MF |
|---|---|---|
| Matrix Factorization | 1.561 | - |
| Deep-In FM | **0.480** | **69.2** % |
| DeepCoNN | 1.441 | 19.6 % |

art for predicting customer ratings when textual reviews are available. For Deep-In FM we use the same feature extraction function (see Appendix A.3 for details) used by DeepCoNN. We evaluate on the Yelp dataset[6], which consists of 4.7 million reviews of restaurants. For each user-item pair, DeepCoNN concatenates the text from all reviews for that item and all reviews by that user. The concatenated text is fed into a feature extraction function followed by a factorization machine. In contrast, for Structured Factorization, we build 3 feature groups: item identifiers (in this case, restaurants), users and review text.

Table 4 compares our methods to DeepConn's published results because a public implementation is not available. We see that Structured Deep-In Factorization Machine provides a large performance increase when comparing the reported improvement, over Matrix Factorization, of the mean squared error. Our approach is more general, and we claim that it is also more efficient. Since DeepCoNN concatenates text, when the average reviews per user is $\bar{n}_u$ and reviews per item is $\bar{n}_i$, each text is duplicated on average $\bar{n}_i \times \bar{n}_u$ times per training epoch. In contrast, for Deep-In FM each review is seen only once per epoch. Thus it can be 1-2 orders of magnitude more efficient for datasets where $\bar{n}_i \times \bar{n}_u$ is large.

## 5 CONCLUSION

We present a general purpose method for factorizing large feature sets; we demonstrate it in several applications, such as using text to enable prediction for unseen items and circumvent the cold-start problem. Future work may soften our requirement of domain knowledge—in general, our methods require feature groups and feature extraction functions defined by experts. We did not pursue an exhaustive comparison with previously published methods; for example, there are other algorithms that rely on Bayesian optimization (De Campos et al., 2010) to infer the item embeddings from text which we did not benchmark. Although we apply our methods on six datasets altogether, further experimentation may be able to situate under which conditions our methods are effective.

Our methods generalize previously published single-purpose neural networks. For example, TagSpace (Weston et al., 2014) is a very successful method, but it is limited to a single textual feature. With the correct feature extraction function, Structured Deep-In Factorization Machine can be used to implement a TagSpace model.

Compared to previous general-purpose approaches, our work makes less assumptions about the training data and allows more flexibility. We provide evidence that the factorization hypothesis may be too restrictive—when relaxed we see higher predictive accuracy with a dramatic improvement of training speed. We show experimental results outperforming an algorithm specifically designed for text—even when using the same feature extraction CNN. This suggests that the need for ad-hoc networks should be situated in relationship to the improvements over a general-purpose method. To the extent of our knowledge, our work is the first to propose a general purpose factorization algorithm that enables efficient inference on arbitrary feature sets.

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

# A  APPENDIX: REPRODUCIBILITY DETAILS

## A.1  HYPER-PARAMETER SETUP FOR STRUCTURED DEEP-OUT FACTORIZATION MACHINE EXPERIMENTS

For the courses dataset, our experimentation strategy killed experiments that took too long. In practice, that means that some shallow experiments with batch size of 5,000 observations were not executed.

Table 5: Grid search hyper-parameters

| Dataset | Mini-batch size | Embedding dimension | Deep-Out Activation |
|---|---|---|---|
| Subscription | 20,000; 40,000 | 5; 10; 20 | ReLU |
| Airline | 1,000; 10,000 | 2; 5; 10; 20 | linear |
| Courses | 5,000; 10,000; 30,000; 40,000 | 30;50;70 | linear |

## A.2  FEATURE EXTRACTION NETWORK FOR NATURAL LANGUAGE

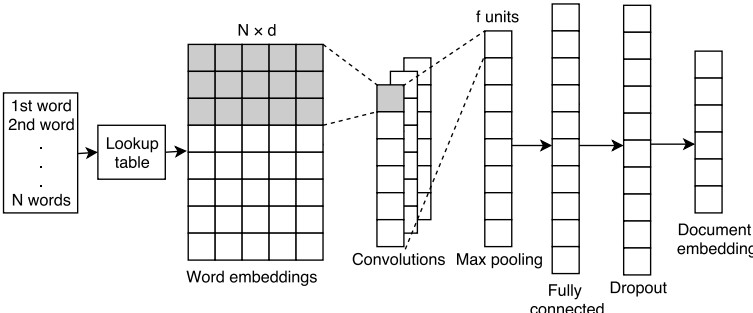

Figure 3: Feature extraction network used for labelling tasks. We use f=1000 convolutional filters each of width 3 (words)

Here we describe the details of the feature extraction function $\phi$ used in our experiments for labelling in §4.2.1 and §4.2.2. An overview of the network is given in Fig. 3. We choose the most popular words of each dataset to build a vocabulary of size $n$, and convert the words of each document to a sequence of length $t$ of one-hot encodings of the input words. If the input text is shorter than $t$, then we pad it with zeros; if the text is longer, we truncate it by discarding the trailing words. Therefore, for a vocabulary size $n$, the input has dimensions $t \times n$, and this matrix is then passed through the following layers:

1. We use an embedding layer to assign a $d$-dimensional vector to each word in the input passage of text. This is done through a $d \times n$-dimensional lookup table, which results in an $t \times d$ matrix.

2. We extract features from the embeddings with functions called *convolutional filters* (LeCun et al., 1998) (also called feature maps). A convolutional filter is simply a matrix learned from an input. We learn $f$ filters that are applied on groups of $m$ adjacent word embeddings, thus each of our filters is a $d \times m$ matrix of learned parameters. Filters are applied by computing the element-wise dot product of the filter along a sliding window of the entire input. The resulting output for each filter is a vector of length $t - m + 1$. We also apply a ReLU activation to the output of each filter.

3. Consider the case of inputs of different lengths. For very short texts, the output of the filters will be mostly zero since the input is zero-padded. To enforce learning from the features of the text, and not just its length we apply a function called 1-max pooling to the output of the filters: from the $t - m + 1$ output vector of each filter, we select the maximum value. This yields a vector of length $F$, a representation of the passage which is independent of its length.

4. We learn higher-level features from the convolutional filters. For this, we use a fully connected layer with $p$ units and a ReLU activation,

5. During training (not in inference), we prevent the units from co-adapting too much with a dropout layer (Srivastava et al., 2014). Dropout is a form of regularization that for each mini-batch randomly drops a specified percentage of units.

6. the final embedding for $x_j$ (that is used in the factorization) is computed by a dense layer with $r$ output units and an activation function, where $r$ is the embedding size of our indexable items.

We set the maximum vocabulary size $n$ to 100,000 words, and input embedding size $d$ to 50 for all experiments. We initialize the input word embeddings and the label embeddings using Word2Vec(Mikolov et al., 2013) We have have not evaluated multiple architectures or hyper-parameter settings and obtain good results on diverse datasets with the same architecture, which was designed followed recommendations from a large scale evaluation of CNN hyper parameters(Zhang & Wallace, 2015). We set the number of convolutional filters $f$ to 1,000, and the dropout rate to 0.1. The maximum sequence length $t$ was chosen according to the typical document length (5,000 words for the concept dataset , 400 for the skills and 350 for CiteULike), the embedding size was set to $r = 100$ for concepts, and $r = 200$ for skills. For the CTR dataset since we use very small values of $r$, due to the tendency of the ReLU units to'die' during training (output zero for all examples), which can have a significant impact, we used instead PReLU activations (He et al., 2015) for the final layer, since they do not suffer from this issue.

## A.3 FEATURE EXTRACTION FOR DEEPCONN COMPARISON

The CNN architecture used for DeepCoNN (Zheng et al., 2017) is similar to that described above in Appenix A.2. It consists of a word embedding lookup table, convolutional layer, 1-max pooling and a fully connected layer. We use the hyper-parameters that the authors report as best - 100 convolution filters and 50 units for the fully connected layer. We set the word embedding size to 100, the vocabulary size to 100,000 and the maximum document length to 250.

## A.4 HYPER-PARAMETERS FOR CTR

To compare Deep-In FM with Collaborative Topic Regression, we choose the embedding size $r \in \{5, 10, 15\}$ for which CTR performs best. The other parameters were set to those reported by the authors as best. The results are shown in Table 6.

Table 6: Tuning embedding size for CTR

|                                            | r=5    | r=10   | r=15   | Time (mins.) |
| ------------------------------------------ | ------ | ------ | ------ | ------------ |
| Matrix Factorization                       | 0.8723 | 0.8911 | 0.9046 | **1**        |
| Structured Deep-In Factorization Machine   | **0.9081** | **0.9303** | **0.9401** | 133    |
| Collaborative Topic Regression             | 0.8763 | 0.9234 | 0.9356 | 1425         |

