# OpenReview forum: "Structured Deep Factorization Machine: Towards General-Purpose Architectures"
_ICLR.cc/2018/Conference — Reject_

### Official Review · AnonReviewer3 · 2017-11-25
**proposes a model that is equivalent to known work**

**Rating:** 3
**Confidence:** 5

**Review:**

This paper proposes to improve time complexity of factorization machine. Unfortunately, the paper's claim that FM's time complexity is quadratic to feature size is wrong. Specifically, the dot product can be computed as (which is linear to feature size)

(\sum x_i \beta_i)^T (\sum x_i \beta_i) - \sum_i x_i^2 beta_i^T beta_i

The projection of feature group into one embedded space proposed in the paper can be viewed as another form of representing the same model when group equals one. When the number of feature groups do not equal one, they correspond to field aware factorization machine(FFM)

---

### Official Review · AnonReviewer2 · 2017-11-30
**Novel model for Collaborative filtering. Seems reasonable overall. Empirical study would be more convincing by including classic recsys datasets˚.**

**Rating:** 4
**Confidence:** 5

**Review:**

The authors introduce a novel novel for collaborative filtering. The proposed model combines some of the strengths of factorization machines and of polynomial regression. Another way to understand this model is that it's a feed forward neural network with a specific connection structure (i.e., not fully connected).

The paper is well written overall and relatively easy to understand. The study seems fairly thorough (both vanilla and cold-start experiments are reported).

Overall the paper feels a little bit incomplete . This is particularly apparent in the empirical study. Given the somewhat limited novelty of the model the potential impact of this work relies on more convincing experimental results. Here are some suggestions about how to achieve that:

1) Methodically report results for MF, FM, CTR (when meaningful), other strong baselines (maybe SLIM?) and all your methods for all datasets.

2) Report results on well-known CF datasets. Movielens comes to mind.

3) Shed some light on some of the poor CTR results (last paragraph of Section 4.2.2)

4) Explore the models and shed some lights on where the gains are coming from.


Minor:

- How do you deal with unobserved preferences in the implicit case?

- I found the idea of Figure 1 very good but in its current form I didn't find it particularly insightful (these "clouds" are hard to interpret).

- It may also be worth adding this reference when discussing neural factorization:
http://www.cs.toronto.edu/~mvolkovs/nips2017_deepcf.pdf

---

### Official Review · AnonReviewer1 · 2017-12-07
**context and contribution of this work is not very clear to me**

**Rating:** 4
**Confidence:** 3

**Review:**

This paper presents a method for matrix factorization using DNNs. The suggestion is to make the factorization machine (eqn 1) deep, by grouping the features meaningfully (eqn 5), extracting nonlinear features from original inputs (deep-in, eqn 8), and adding additional nonlinearity after computing pairwise interactions (deep-out, eqn 7). From the methodology point of view, such extensions are relatively straightforward. As an example, from the experimental results, it seems the grouping of features is done mostly with domain knowledge (e.g., months of year) and not learned automatically. The authors claim the proposed method can circumvent the cold-start problem, and presented some experimental results on recommendation systems with text features.

While the application problems look quite interesting, in my opinion, the paper needs to make the context and contribution clearer. In particular, there is a huge literature in collaborative filtering, and I believe there is by now sufficient work on collaborative filtering with input features (and possibly dealing with the cold-start problem). I think this paper does not connect very well with that literature. When reading it, at times I felt the main purpose of this paper is to solve the application problems presented in experimental results, instead of proposing a general framework. I suggest the authors to demonstrate their method on some well-known datasets (e.g., MovieLens, Netflix), to give the readers an idea if the proposed method is indeed advantageous over more classical methods, or if the success of this paper is mostly due to clever processing of text features using DNNs.

Some detailed comments:
1. eqn 4 does not indicate any rank-r factors.
2. some statements do not seem straightforward/justified to me:
    -- the paper uses the word "inference" several times without definition
    -- "if we were interested in interpreting the parameters, we could constrain w to be non-negative ... ". Is this easy to do, and can the authors demonstrate this in their experiments and show interpretable examples?
    -- "Note that if the dot product is replaced with a neural function, fast inference for cold-start ...".
3. the experimental setup seems quite unusual to me: "since we only observe positive labels, for such tasks in the test set we sample a labels according to the label frequency". This seems very problematic if most of the entries are not observed. Why cannot you use the typical evaluation procedure for collaborative filtering, where you hide some known entries during model training, and evaluate on these entries during test?

---

### Author Response · Authors · 2018-01-05
**Withdrawing paper**

Dear reviewers,

Thank you for your very insightful comments. We are withdrawing this paper and using some of these results to supplement a different paper.

We are not clicking on "withdrawing" paper yet, as this submission would be de-anonymized immediately.

Thanks,
Author

---

### Decision · Program_Chairs · 2018-01-29
**ICLR 2018 Conference Acceptance Decision**

**Decision:**

Reject

**Comment:**

This paper has been withdrawn by the authors.